# COLA: Towards Efficient Multi-Objective Reinforcement Learning with Conflict Objective Regularization in Latent Space

**Pengyi Li\*, Hongyao Tang†, Yifu Yuan, Jianye Hao†, Zibin Dong, Yan Zheng**
College of Intelligence and Computing, Tianjin University

## Abstract

Many real-world control problems require continual policy adjustments to balance multiple objectives, which requires the acquisition of high-quality policies to cover diverse preferences. Multi-Objective Reinforcement Learning (MORL) provides a general framework to solve such problems. However, current MORL methods suffer from high sample complexity, primarily due to the neglect of efficient knowledge sharing and conflicts in optimization with different preferences. To this end, this paper introduces a novel framework, Conflict Objective Regularization in Latent Space (**COLA**). To enable efficient knowledge sharing, COLA establishes a shared latent representation space for common knowledge, which can avoid redundant learning under different preferences. Besides, COLA introduces a regularization term for the value function to mitigate the negative effects of conflicting preferences on the value function approximation, thereby improving the accuracy of value estimation. The experimental results across various multi-objective continuous control tasks demonstrate the significant superiority of COLA over the state-of-the-art MORL baselines. Code is available at https://github.com/yeshenpy/COLA.

## 1 Introduction

Real-world problems often involve multiple performance metrics, where conflicts among these metrics make it infeasible to maximize all metrics simultaneously with a single policy. This necessitates a trade-off among different metrics, requiring the learning of multiple high-quality policies to meet diverse preferences [1–3]. For instance, in robot control problems, performance metrics may include both forward speed and energy consumption. In such cases, two reward signals are returned, and multiple optimal policies exist based on different trade-offs between the two metrics [4]. Learning these policies enables us to control the robot more flexibly [5]. For example, initially, one might prefer a policy that prioritizes speed without considering energy consumption. Still, as energy loss accumulates, a preference may shift towards an energy-efficient policy to extend running time.

A popular approach to addressing the multi-objective control problems involves leveraging Multi-Objective Reinforcement Learning (MORL) to acquire policies that meet human preferences [6, 7]. Current MORL algorithms can be roughly categorized into three categories: Single-policy methods, Multi-policy methods, and General policy methods. Single-policy methods directly transform multi-objective problems into single-objective problems and solve them using standard RL [6, 8, 9]. These methods typically require expert knowledge to predefine preferences, and the obtained policies are difficult to generalize to unknown preferences. Multi-policy methods seek a set of policies to cover optimal solutions under diverse preferences, which often require a substantial amount of sample

---

\*Contact me at lipengyi@tju.edu.cn
†Corresponding authors: Hongyao Tang (tanghongyao@tju.edu.cn) & Jianye Hao (jianye.hao@tju.edu.cn)

39th Conference on Neural Information Processing Systems (NeurIPS 2025).

cost to construct high-quality policies [10–12]. General policy methods aim to construct a single network with preferences as the condition to cover all policies, which have higher sample efficiency compared to the above two categories due to the implicit knowledge sharing [13–15].

However, the general policy methods still require a large number of samples to learn high-quality policies. We summarize two challenges. 1) *Inefficiency of Knowledge Sharing*: Although using a single network allows implicit knowledge sharing, policies under different preferences need to redundantly learn common knowledge, resulting in inefficient learning. 2) *Conflict of Optimization Directions*: Different preferences may conflict in optimization directions. When using a shared network to approximate the value of a specific preference, it may compromise the approximation of the values of other preferences, resulting in a tug-of-war during optimization. These conflicts impede the accurate approximation of the value function to distinct preferences, thereby resulting in suboptimal policies.

To address these challenges, we propose a novel framework named Conflict Objective Regularization in Latent Space (COLA). The core insight of COLA is to *seek commonalities while accommodating differences*, i.e., extracting and sharing the common knowledge required for learning under various preferences, while mitigating interference between different preferences when their optimization directions conflict. COLA comprises two main components: 1) To solve the problem of *Inefficiency of Knowledge Sharing*, we introduce the Objective-agnostic Latent Dynamics Model (OADM), which utilizes underlying dynamics transitions in the environment to extract a universal knowledge representation through temporal latent consistency. The knowledge extracted by OADM provides more compact general information of dynamics and is essential for learning policies under all preferences. Subsequent multi-objective optimization based on the latent representation constructed by OADM can thus be more efficient. 2) To address the problem of *Conflict of Optimization Directions*, COLA introduces Conflict Objective Regularization (COR). When the optimization direction of the current preference conflicts with the optimization direction of other preferences, COR constrains the value estimates for other preferences from deviating away from the original value estimates. This minimizes negative impacts (or interference) between optimizations for different preferences, allowing the value function to simultaneously maintain value estimates for multiple preferences. The experiments in a range of multi-objective control tasks show that COLA significantly outperforms current state-of-the-art MORL algorithms across various general metrics, achieving higher sample efficiency and ultimate performance.

## 2 Background

### 2.1 Multi-Objective Markov Decision Process

Consider a Multi-Objective Markov Decision Process (MOMDP) [16], defined by a tuple $\langle \mathcal{S}, \mathcal{A}, \mathcal{P}, \boldsymbol{r}, \gamma, \Omega, f_\Omega \rangle$ with state space $\mathcal{S}$, action space $\mathcal{A}$, transition function $\mathcal{P}(s, a)$, the vector-valued reward function $\boldsymbol{r}$ with $d$ objectives, the discount factor $\gamma \in (0, 1)$, the preference space $\Omega$ and the scalarization function $f_\Omega$ which produces a scalar utility using preference $\boldsymbol{\omega} \in \Omega$. In our work, we use a linear preference function, i.e., $f_{\boldsymbol{\omega}}(\boldsymbol{r}(s, a)) = \boldsymbol{\omega}^\top \boldsymbol{r}(s, a)$, which is commonly adopted in the MORL works [13, 14, 17]. When the number of objectives is $d = 1$, the MOMDP simplifies to a standard MDP. If multiple objectives ($d > 1$) and all possible returns are considered, a set of Pareto optimal solutions called Pareto Frontier (PF) can be obtained. A policy $\pi$ is Pareto Optimal when no other policy $\pi'$ can increase its expected return in any objective without decreasing the expected return in any other objective. In continuous control tasks, obtaining the PF is an NP-hard problem [4]. Therefore, the goal of MORL is to obtain an approximation of the PF.

In this paper, we focus on the general policy methods. Therefore, our objective is to construct a general policy $\pi(a|s, \boldsymbol{\omega})$ [18, 19] that, given a state $s \in \mathcal{S}$, can generalize across the entire space of preferences by adjusting the preference $\boldsymbol{\omega} \in \Omega$.

### 2.2 Envelope Soft Actor Critic

We introduce a foundational MORL algorithm, which integrates Envelope Q-Learning with SAC. We refer to this algorithm as Envelope SAC [14, 20], which serves as the foundational algorithm for our work. Specifically, the multi-objective Bellman equation for Envelope SAC can be formalized

as follows:

$$\mathcal{L}_{\boldsymbol{Q}} = \mathbb{E}_{D, \boldsymbol{\omega} \sim \Omega} \left[ \boldsymbol{\omega}^\top \left( \boldsymbol{Q}_\phi(s, a, \boldsymbol{\omega}) - \left( \boldsymbol{r} + \gamma \boldsymbol{V}_{\bar{\xi}}(s', \boldsymbol{\omega}) \right) \right)^2 \right]$$
$$\boldsymbol{V}_\xi(s, \boldsymbol{\omega}) = \mathbb{E} \left[ \boldsymbol{Q}_\phi \left( s, \pi_\theta(a|s, \omega), \boldsymbol{\omega} \right) - \alpha \log \pi_\theta \left( a|s, \boldsymbol{\omega} \right) \mathbf{1}_d \right],$$

(1)

where $(s, a, \boldsymbol{r}, s')$ is sampled from the replay buffer $D$, $\phi, \theta, \xi$ (or $\bar{\xi}$) are the parameters for corresponding networks (or the target network), and $\mathbf{1}_d$ denotes a $d$-dimensional vector of all ones. To ensure that the samples contribute to the optimization of all preferences, $\boldsymbol{\omega}$ is randomly sampled from the preference space $\Omega$ rather than using the preferences applied during the interaction. The general policy for different preferences is optimized through the following equation:

$$\mathcal{L}_\pi(\boldsymbol{\omega}) = \mathbb{E} \left[ \sup_{\boldsymbol{\omega}' \in \Omega} \left\{ \alpha \log \left( \pi_\theta(a|s, \boldsymbol{\omega}) \right) - \boldsymbol{\omega}^\top \boldsymbol{Q} \left( s, a, \boldsymbol{\omega}' \right) \right\} \right],$$

(2)

where $a$ is sampled from $\pi_\theta$. We follow the structures of SAC, maintaining two Q-functions to alleviate overestimation issues and adjusting $\alpha$ in a learning manner.

## 3 COLA Framework

This section introduces our framework Conflict Objective Regularization in Latent Space (COLA). We begin by introducing how to construct a latent space by Objective-agnostic Latent Dynamics Model for efficient knowledge sharing. Subsequently, we elaborate on how to mitigate the impact of conflicting preferences on value function approximation through Conflict Objective Regularization. Finally, we provide an overview of the COLA framework.

### 3.1 Objective-agnostic Latent Dynamics Model

In complex multi-objective continuous control tasks, optimizing a policy or set of policies to cover the entire preference space typically requires a significant sample cost. To enhance sample efficiency, general policy methods employ shared networks and integrate preferences as conditional inputs to facilitate implicit knowledge sharing across various preferences. However, the lack of efficient extraction and utilization of general knowledge limits the efficiency of this learning approach.

To solve the problem, we propose the Objective-agnostic Latent Dynamics Model (OADM) to extract and share the representations of common knowledge in the underlying environment dynamics transitions. Our key insight is that the dynamic transition information underlies robot motion and must be mastered, regardless of the preference policies for robots. By explicitly extracting and sharing the representations, the policies avoid redundant learning of the common knowledge for different preferences, and a more compact and beneficial latent space is constructed, making multi-objective optimization more efficient.

The architectural diagram of OADM is depicted in Figure 1. Specifically, the OADM consists of two components: a state encoder $\mathbf{E}_\psi$ and a transition model $\mathbf{D}_\psi$. We formalize the components as follows:

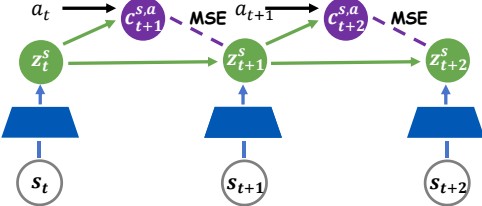

Encoder : $\quad z_t = \mathbf{E}_\psi(s_t)$
Transition : $c_{t+1} = \mathbf{D}_\psi(z_t, a_t)$

In OADM, the encoder maps state $s_t$ to the latent representation $z_t$. The transition model takes the latent representation $z_t$ and $a_t$ as inputs and output $c_{t+1}$. To extract the consistent representations in dynamic transitions, OADM leverages the temporal consistency loss to make $c_{t+1}$ predict the latent representation $z_{t+1}$ of the next state $s_{t+1}$. The loss is defined as follows:

Figure 1: The conceptual illustration of Objective-agnostic Latent Dynamics Model (OADM). We construct consistent state representations $z_t^s$ and state-action representations $c_{t+1}^{s,a}$ in latent space for extracting common dynamics information.

$$\mathcal{L}_{\texttt{OADM}}(\psi; \tau) = \mathbb{E}_D \left[ \left( \mathbf{E}_\psi(s_{t+1}) - \mathbf{D}_\psi(z_t, a_t) \right)^2 \right],$$

(3)

where $(s_t, a_t, s_{t+1})$ is sampled from the replay buffer $D$.

The temporal consistency loss is used in several previous methods [21–26], which are predominantly tailored for single-objective optimization tasks. In addition to capturing dynamics, these methods inject task-specific information by maximizing the value function or predicting rewards. However, when applied to multi-objective optimization, this approach may introduce potential conflicts, particularly in the optimization of conflicting objectives. Therefore, in COLA, we remove the influence of different preference values and rewards on OADM. This intentional separation allows us to concentrate specifically on providing objective-agnostic state transition information.

For the multi-objective optimization process, the main distinction lies in the optimization being performed in the more compact latent space constructed by OADM, rather than in the original state space. The latent policy optimization process is depicted in Figure 2, which consists of two parts: multi-objective value function approximation and multi-objective policy optimization. Specifically, to optimize more efficiently, the value function takes the latent state representation $z_t$, the latent state-action representation $c_{t+1}$, the action $a_t$, and the preference $\omega$ as inputs. The original state information $s_t$ can be optional. However, in the experiments, we observe that discarding the original information $s_t$ sometimes results in unstable policy updates, possibly due to real-time updates of OADM. Therefore, during the learning process,

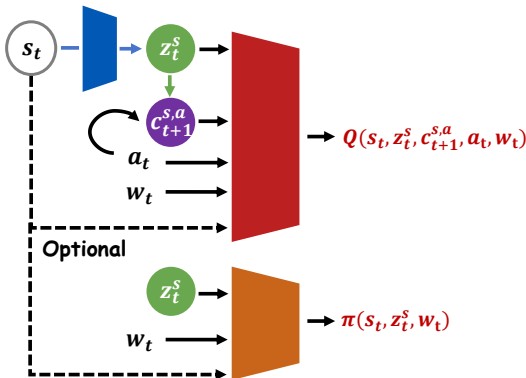

Figure 2: Multi-objective policy optimization in latent space.

we also take the stable input $s_t$ to mitigate the problem. The formulation of the optimization loss $\mathcal{L}_Q$ can be rewritten as follows:

$$\mathcal{L}_Q = \mathbb{E}_D \left[ \boldsymbol{\omega}^\top \left( \boldsymbol{Q}_\phi(s_t, z_t, c_{t+1}, a_t, \boldsymbol{\omega}) - y \right)^2 \right], \tag{4}$$

where $y = \boldsymbol{r}(s_t, a_t) + \gamma \boldsymbol{V}_{\bar{\xi}} \left( s_{t+1}, z_{t+1}, \boldsymbol{\omega} \right)$. The policy $\pi$ makes decisions by taking $s_t$, $z_t$ and $\boldsymbol{\omega}$ as inputs. The loss $\mathcal{L}_\pi$ of policy is formulated as follows:

$$\mathcal{L}_\pi(\boldsymbol{\omega}) = \mathbb{E}_D \left[ \mathbb{E}_{a_t \sim \pi} \left[ \sup_{\boldsymbol{\omega}' \in \Omega} \left\{ \alpha \log \left( \pi_\theta(a_t \mid s_t, z_t, \boldsymbol{\omega}) \right) - \boldsymbol{\omega}^\top \boldsymbol{Q} \left( s_t, z_t, \mathbf{D}(z_t, a_t), a_t, \boldsymbol{\omega}' \right) \right\} \right] \right]. \tag{5}$$

With efficient representation and sharing of common knowledge, multi-objective value function and policy optimization can become more efficient.

## 3.2 Conflict Objective Regularization

The multi-objective value network is crucial for providing policy gradient guidance under various preferences. In current general policy methods, a shared architecture is employed for the value network. Leveraging the shared architecture facilitates task-related knowledge sharing among similar preferences, leading to more efficient learning. However, in multi-objective optimization, numerous conflicting preferences exist. Approximating the value for one preference can have negative impacts on the value estimates for other potentially conflicting preferences. In essence, this is a manifestation of Neural Network Churn [27, 28] in the context of multi-objective optimization. To this end, naturally we aim to study: *How can we mitigate the negative impact of conflicting preferences?*

To solve the problem, we propose the Conflict Objective Regularization Loss (COR) to mitigate the negative effects of optimization direction conflicts among preferences. When the value function approximates the values of preference $\boldsymbol{\omega}_i$, COR aims to minimize the value estimate variation of random preference $\boldsymbol{\omega}_j$ if the optimization directions of the value function under preferences $\boldsymbol{\omega}_i$ and $\boldsymbol{\omega}_j$ conflict. This enhances the stability of value function learning, preventing drastic changes in the estimated values of conflicting preferences, thus preserving effective policy gradient guidance under different preferences.

To implement COR, we first need to establish a metric for the degree of conflict. Concretely, we measure the stiffness $\rho\left(\boldsymbol{\omega}_i, \boldsymbol{\omega}_j\right)$ [29–31] of the value network between preferences $\boldsymbol{\omega}_i$ and $\boldsymbol{\omega}_j$, which

---

**Algorithm 1** COLA Algorithm

---

1: **Initialize:** a replay buffer $\mathcal{D}$, the Objective-agnostic Latent Dynamics Model with encoder $\mathbf{E}_\psi$ and transition model $\mathbf{D}_\psi$, the MO policy $\pi_\theta$, the MO value function $\mathbf{Q}_\phi$, $\mathbf{V}_\xi$.
2: **repeat**
3:    # Interaction for experiences
4:    Sample a preference $\boldsymbol{\omega_i}$ from $\Omega$
5:    The MO policy interacts with the environment based on $\boldsymbol{\omega_i}$
6:    Store the transitions $\{s, a, s', r, \text{done}\}$ to $\mathcal{D}$
7:    # Optimize the OADM
8:    Optimize OADM with experiences in $\mathcal{D}$ {see Eq. 3}
9:    # MO Learning in the latent space
10:    **Optimize the Critic:**
11:    Sample experiences to calculate the COR loss with $\boldsymbol{\omega_i}$ and a random sampled $\boldsymbol{\omega_j}$ {see Eq. 7}
12:    Optimize $\boldsymbol{Q}$ to approximate the values under $\boldsymbol{\omega_i}$ {see Eq. 8}
13:    **Optimize the Policy:** Optimize $\pi$ under $\boldsymbol{\omega_i}$ {see Eq. 5}
14: **until** reaching maximum training steps

---

is defined by:

$$\rho\left(\boldsymbol{\omega_i}, \boldsymbol{\omega_j}\right) = \frac{\nabla_\phi \mathcal{L}_{\boldsymbol{Q}}(\phi; \boldsymbol{\omega_i}))^\top \nabla_\phi \mathcal{L}_{\boldsymbol{Q}}\left(\phi; \boldsymbol{\omega_j}\right)}{\left\|\nabla_\phi \mathcal{L}_{\boldsymbol{Q}}(\phi; \boldsymbol{\omega_i})\right\|_2 \left\|\nabla_\phi \mathcal{L}_{\boldsymbol{Q}}\left(\phi; \boldsymbol{\omega_j}\right)\right\|_2}. \tag{6}$$

Low stiffness indicates that updating the value network parameters toward minimizing $\mathcal{L}_{\boldsymbol{Q}}$ for preference $\boldsymbol{\omega_i}$ will have a negative effect on the minimization of $\mathcal{L}_{\boldsymbol{Q}}$ for preference $\boldsymbol{\omega_j}$. Meanwhile, a higher stiffness indicates a more consistent optimization direction for the value network parameters between $\boldsymbol{\omega_i}$ and $\boldsymbol{\omega_j}$. We set a conflict boundary, denoted as $c$. When the stiffness $\rho\left(\boldsymbol{\omega_i}, \boldsymbol{\omega_j}\right)$ is greater than or equal to $c$, we consider that the optimization directions of the value functions under preferences $\boldsymbol{\omega_i}$ and $\boldsymbol{\omega_j}$ are relatively consistent, mutually reinforcing each other. However, when $\rho\left(\boldsymbol{\omega_i}, \boldsymbol{\omega_j}\right)$ is less than $c$, we assume that there is a conflict in the optimization directions between $\omega_i$ and $\boldsymbol{\omega_j}$. Additionally, a larger value of c-$\rho\left(\boldsymbol{\omega_i}, \boldsymbol{\omega_j}\right)$ indicates a more severe conflict.

If $\rho\left(\boldsymbol{\omega_i}, \boldsymbol{\omega_j}\right) - c < 0$, we introduce the COR loss to the value function under the preference $\boldsymbol{\omega_j}$ to ensure that the optimization under $\boldsymbol{\omega_i}$ does not significantly compromise the value estimation under $\boldsymbol{\omega_j}$. The COR loss can be formalized as follows:

$$\mathcal{L}_{\text{COR}} = \mathbb{E}_D\left[\underbrace{\max(c - \rho\left(\boldsymbol{\omega_i}, \boldsymbol{\omega_j}\right), 0)}_{\lambda} \cdot \underbrace{(\boldsymbol{Q'}(\boldsymbol{\omega_j}) - \boldsymbol{Q}(\boldsymbol{\omega_j}))^2}_{\Delta(\boldsymbol{\omega_j})}\right], \tag{7}$$

where the term $\lambda$ represents the degree of conflict, with a larger value indicating a greater conflict, consequently, the regularization term's constraining effect will be stronger. The term $\Delta(\boldsymbol{\omega_j})$ represents the magnitude of the change in $\boldsymbol{Q}$. $\boldsymbol{Q'}$ is the value function of $\boldsymbol{Q}$ optimized based on the preference $\boldsymbol{\omega_i}$, i.e., $\boldsymbol{Q} \xrightarrow{\boldsymbol{\omega_i}} \boldsymbol{Q'}$. Minimizing $\Delta(\boldsymbol{\omega_j})$ helps mitigate the impact on the value estimation under $\boldsymbol{\omega_j}$. In the practical implementation, obtaining the optimized value function $\boldsymbol{Q'}$ for the next step to optimize the current parameters is challenging. This can be addressed by leveraging the MAML method [32]. However, it introduces significant additional computational overhead. To simplify this process, we employ a surrogate approach by maintaining an old Q network from the previous step, denoted as $\boldsymbol{Q}_{\text{old}}$, and using the squared difference between the current $\boldsymbol{Q}(\boldsymbol{\omega_j})$ and the previous step's $\boldsymbol{Q}_{\text{old}}(\boldsymbol{\omega_j})$ as a replacement for $\Delta(\boldsymbol{\omega_j})$. Finally, the value function loss of COLA is defined as follows:

$$\mathcal{L}_{\text{COLA}} = \mathcal{L}_{\boldsymbol{Q}} + \beta \cdot \mathcal{L}_{\text{COR}}, \tag{8}$$

where $\beta$ is the hyperparameter to balance the impact of COR loss. Through $\mathcal{L}_{\text{COLA}}$, we can reduce the negative impact on the value functions of other preferences when optimizing any preference $\boldsymbol{\omega_i}$, thereby enhancing the ability of the value function to maintain values for different preferences.

### 3.3 COLA Algorithm

We provide the pseudo-code of COLA in Algorithm 1. In each iteration, the algorithm proceeds across three phases (denoted by blue). First, we randomly sample a preference $\boldsymbol{\omega_i}$ from the preference space $\Omega$ and interact with the environment based on $\pi$ and $\boldsymbol{\omega_i}$, the interaction experiences

Table 1: Comparison of COLA with other MORL algorithms across ten tasks in terms of the three metrics. All algorithms are trained for 3 million environment steps. The reference point is set to the origin for HV calculation. COLA achieves remarkable superiority over other algorithms in all metrics with an equivalent number of samples.

| Tasks | Metric | Envelope SAC | PGMORL | Q-Pensieve | CAPQL | COLA (Ours) |
|---|---|---|---|---|---|---|
| HalfCheetah-2d | HV ($\times 10^6$) | $5.54 \pm 0.47$ | $5.13 \pm 0.06$ | $5.66 \pm 0.16$ | $5.84 \pm 0.14$ | $\mathbf{7.91 \pm 0.06}$ |
| | UT ($\times 10^3$) | $2.19 \pm 0.11$ | $2.14 \pm 0.03$ | $2.22 \pm 0.06$ | $2.24 \pm 0.03$ | $\mathbf{2.90 \pm 0.07}$ |
| | ED | $0.40 \pm 0.04$ | $0.34 \pm 0.00$ | $0.37 \pm 0.02$ | $0.50 \pm 0.00$ | $\mathbf{0.89 \pm 0.00}$ |
| Hopper-2d | HV ($\times 10^7$) | $2.00 \pm 0.10$ | $1.90 \pm 0.07$ | $2.06 \pm 0.08$ | $1.65 \pm 0.07$ | $\mathbf{2.11 \pm 0.03}$ |
| | UT ($\times 10^3$) | $4.15 \pm 0.09$ | $4.12 \pm 0.07$ | $4.21 \pm 0.07$ | $3.95 \pm 0.09$ | $\mathbf{4.28 \pm 0.02}$ |
| | ED | $0.61 \pm 0.00$ | $0.23 \pm 0.00$ | $0.69 \pm 0.01$ | $0.04 \pm 0.00$ | $\mathbf{0.91 \pm 0.00}$ |
| Ant-2d | HV ($\times 10^6$) | $6.90 \pm 0.06$ | $5.40 \pm 0.62$ | $6.93 \pm 0.16$ | $7.07 \pm 0.98$ | $\mathbf{11.33 \pm 2.76}$ |
| | UT ($\times 10^3$) | $2.38 \pm 0.01$ | $2.12 \pm 0.11$ | $2.39 \pm 0.03$ | $2.43 \pm 0.15$ | $\mathbf{3.04 \pm 0.40}$ |
| | ED | $0.44 \pm 0.05$ | $0.00 \pm 0.00$ | $0.62 \pm 0.01$ | $0.54 \pm 0.09$ | $\mathbf{0.90 \pm 0.02}$ |
| Walker-2d | HV ($\times 10^6$) | $4.47 \pm 0.08$ | $4.50 \pm 0.16$ | $4.37 \pm 0.19$ | $3.35 \pm 0.26$ | $\mathbf{4.87 \pm 0.05}$ |
| | UT ($\times 10^3$) | $1.95 \pm 0.03$ | $1.94 \pm 0.05$ | $1.91 \pm 0.04$ | $1.78 \pm 0.05$ | $\mathbf{2.02 \pm 0.03}$ |
| | ED | $0.52 \pm 0.00$ | $0.51 \pm 0.00$ | $0.32 \pm 0.01$ | $0.26 \pm 0.01$ | $\mathbf{0.90 \pm 0.00}$ |
| Hopper-3d | HV ($\times 10^{10}$) | $3.29 \pm 0.27$ | $3.35 \pm 0.07$ | $3.29 \pm 0.20$ | $1.92 \pm 0.23$ | $\mathbf{3.41 \pm 0.04}$ |
| | UT ($\times 10^3$) | $3.21 \pm 0.04$ | $3.21 \pm 0.01$ | $3.20 \pm 0.04$ | $2.70 \pm 0.11$ | $\mathbf{3.22 \pm 0.06}$ |
| | ED | $0.61 \pm 0.03$ | $0.55 \pm 0.00$ | $0.45 \pm 0.03$ | $0.03 \pm 0.00$ | $\mathbf{0.85 \pm 0.01}$ |
| Ant-3d | HV ($\times 10^9$) | $5.76 \pm 0.09$ | $7.80 \pm 0.52$ | $7.32 \pm 2.97$ | $9.88 \pm 0.89$ | $\mathbf{18.58 \pm 0.76}$ |
| | UT ($\times 10^3$) | $1.60 \pm 0.01$ | $1.90 \pm 0.02$ | $1.82 \pm 0.24$ | $2.02 \pm 0.06$ | $\mathbf{2.68 \pm 0.04}$ |
| | ED | $0.13 \pm 0.01$ | $0.39 \pm 0.01$ | $0.31 \pm 0.04$ | $0.72 \pm 0.00$ | $\mathbf{0.95 \pm 0.00}$ |
| Ant-4d | HV ($\times 10^{14}$) | $4.30 \pm 0.75$ | $3.15 \pm 0.11$ | $3.90 \pm 1.23$ | $5.03 \pm 0.12$ | $\mathbf{7.23 \pm 2.84}$ |
| | UT ($\times 10^3$) | $4.53 \pm 0.19$ | $4.19 \pm 0.07$ | $4.36 \pm 0.34$ | $4.71 \pm 0.02$ | $\mathbf{5.17 \pm 0.45}$ |
| | ED | $0.51 \pm 0.01$ | $0.15 \pm 0.02$ | $0.16 \pm 0.02$ | $0.70 \pm 0.00$ | $\mathbf{0.97 \pm 0.00}$ |
| HalfCheetah-5d | HV ($\times 10^{16}$) | $5.24 \pm 2.00$ | $7.52 \pm 0.52$ | $5.03 \pm 2.77$ | $7.56 \pm 2.59$ | $\mathbf{9.15 \pm 1.58}$ |
| | UT ($\times 10^3$) | $2.56 \pm 0.32$ | $2.64 \pm 0.02$ | $2.60 \pm 0.26$ | $2.65 \pm 0.35$ | $\mathbf{3.11 \pm 0.12}$ |
| | ED | $0.39 \pm 0.02$ | $0.56 \pm 0.00$ | $0.46 \pm 0.03$ | $0.48 \pm 0.02$ | $\mathbf{0.61 \pm 0.00}$ |
| Hopper-5d | HV ($\times 10^{16}$) | $7.83 \pm 0.95$ | $9.80 \pm 0.31$ | $8.68 \pm 0.89$ | $3.37 \pm 2.51$ | $\mathbf{10.41 \pm 0.46}$ |
| | UT ($\times 10^3$) | $2.59 \pm 0.05$ | $2.71 \pm 0.01$ | $2.63 \pm 0.02$ | $1.68 \pm 0.05$ | $\mathbf{2.71 \pm 0.01}$ |
| | ED | $0.44 \pm 0.01$ | $0.67 \pm 0.00$ | $0.59 \pm 0.02$ | $0.12 \pm 0.00$ | $\mathbf{0.68 \pm 0.00}$ |
| Ant-5d | HV ($\times 10^{16}$) | $3.25 \pm 0.10$ | $3.12 \pm 0.04$ | $3.22 \pm 0.05$ | $3.53 \pm 0.08$ | $\mathbf{3.56 \pm 0.09}$ |
| | UT ($\times 10^3$) | $2.02 \pm 0.03$ | $2.01 \pm 0.02$ | $2.03 \pm 0.003$ | $2.04 \pm 0.02$ | $\mathbf{2.05 \pm 0.02}$ |
| | ED | $0.54 \pm 0.02$ | $0.50 \pm 0.00$ | $0.50 \pm 0.01$ | $0.39 \pm 0.03$ | $\mathbf{0.57 \pm 0.00}$ |

are then stored into the replay buffer $D$. Subsequently, we optimize the Objective-agnostic Latent Dynamics Model (OADM) based on Eq. 3, extracting dynamic transition representations and constructing a more compact latent space. Within this latent space, we randomly sample a new preference $\omega_i$ to optimize the value function and policy. To calculate COR loss, an additional preference $\omega_j$ is sampled, and the value function is optimized based on our new loss with Eq. 8. Finally, policy improvement is conducted on the policy based on Eq. 5.

## 4 Experiments

This section provides a comprehensive evaluation of COLA through experiments. We first present the experimental setup, followed by performance comparisons and experimental analysis.

### 4.1 Experimental Setting

Our experiments are primarily conducted on multi-objective continuous control tasks proposed by PGMORL [4]. Additionally, we construct more complex multi-objective optimization tasks, such as Ant, Hopper, and HalfCheetah, each with more than two objectives. Detailed information can be found in Appendix B. We compare COLA with several state-of-the-art MORL algorithms, including Envelope [14], PGMORL [4], Q-Pensieve [20], and CAPQL [33]. For a fair comparison, we use the official implementation, fine-tune the hyperparameters and present the best results obtained in each

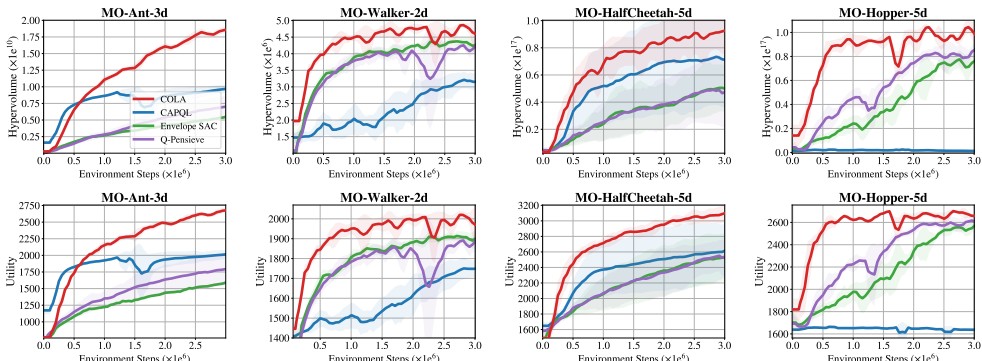

Figure 3: Hypervolume and Expected Utility comparison for continuous control tasks.

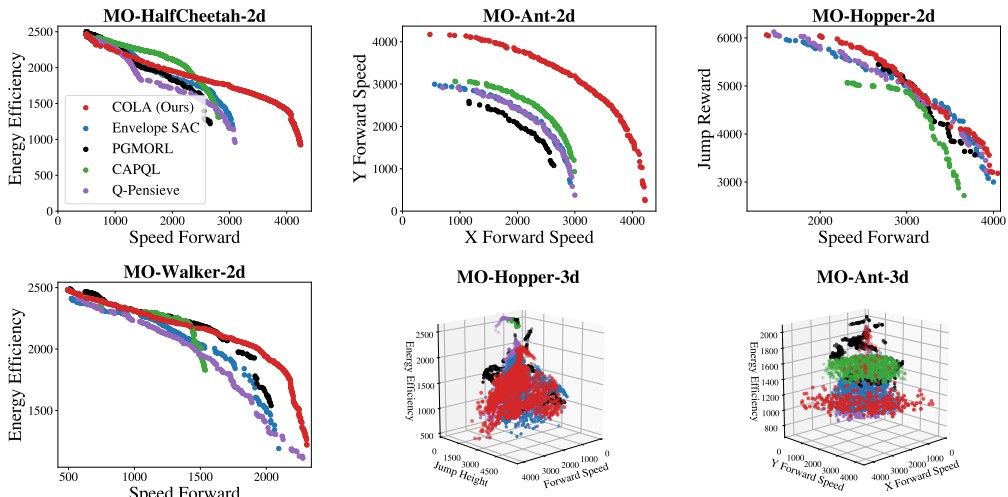

Figure 4: Pareto front comparison for continuous control tasks.

task. It's worth noting that, as the original Envelope is tailored for discrete tasks, we provide its SAC version for comparison in continuous tasks. All algorithms are trained for 3 million environment steps. All statistics are obtained from 5 independent runs and the mean and standard deviation are reported, which is consistent with the setting in previous MORL works. More implementation details can be found in Appendix C.

**Performance Metrics:** We follow the performance evaluation metrics proposed in previous methods [20], which focus on the following metrics:

- **Hypervolume (HV)** is a metric designed to quantify the convergence speed of a Pareto frontier and the diversity of policies within it. HV is defined $\mathrm{HV} := \int_{H(\mathcal{R})} \mathbb{I}\{z \in H(\mathcal{R})\} dz$, where $\mathcal{R}$ is a set of return vectors attained, $r_0 \in \mathbb{R}^d$ is a reference point, $H(\mathcal{R}) := \{z \in \mathbb{R}^d : \exists r \in \mathcal{R}, r_0 \prec z \prec r\}$ and $\mathbb{I}$ is the indicator function.
- **Expected Utility (UT)** serves as a metric to evaluate the overall performance of policies under linear scalarization and is defined as $\mathrm{UT} := \mathbb{E}_{\omega \in \Omega}\left[\max_{r \in \mathcal{R}}(\omega^\top r)\right]$, where $\mathcal{R}$ is a set of return vectors attained.
- **Episodic Dominance (ED)** is defined as $\mathrm{ED}_{1,2} := \mathbb{E}_{\omega \in \Omega}\left[\mathbb{I}\left\{\sum_{t=0}^{T_1} \omega^\top r_t^1 > \sum_{t=0}^{T_2} \omega^\top r_t^2\right\}\right]$, where $r_t^1, r_t^2$ are return vectors, and $T_1, T_2$ are the episode lengths. ED is a metric designed for pairwise comparison of algorithms, particularly suitable for problems where there is a significant difference in returns for different preferences. In such cases, HV and UT may be dominated by a minority of preferences. We calculate ED between algorithms pairwise and report the average ED.

Table 2: Comparison between Envelope and COLA on DMC.

| Tasks | Metric | Envelope | COLA |
|-------|--------|----------|------|
| Cheetah_Run-2d | HV ($\times 10^4$) | $8.61 \pm 0.02$ | $\mathbf{12.71 \pm 0.02}$ |
| Cheetah_Run-2d | UT | $318.80 \pm 29.25$ | $\mathbf{415.98 \pm 30.51}$ |
| Hopper_Hop-2d | HV ($\times 10^2$) | $3.47 \pm 1.60$ | $\mathbf{31.59 \pm 0.14}$ |
| Hopper_Hop-2d | UT | $122.06 \pm 1.93$ | $\mathbf{128.53 \pm 0.78}$ |
| Walker_walk-2d | HV ($\times 10^5$) | $5.89 \pm 3.98$ | $\mathbf{7.73 \pm 3.14}$ |
| Walker_walk-2d | UT | $238.42 \pm 84.43$ | $\mathbf{311.69 \pm 59.90}$ |

Table 3: Comparison between Envelope and COLA on the DeepSea Treasure task.

| DeepSea Treasure | Envelope | COLA |
|------------------|----------|------|
| HV | $799.9 \pm 97.00$ | $\mathbf{1028.93 \pm 3.37}$ |
| UT | $6.323 \pm 0.64$ | $\mathbf{7.94 \pm 0.08}$ |

## 4.2 Performance Evaluation

We evaluate COLA and related strong MORL baselines across ten continuous control tasks. The experimental results shown in Table 1 indicate a significant performance advantage of COLA over other methods. Compared to the second-best approach in different tasks, COLA demonstrates an average improvement of 26.8% in HV and 12% in UT. In pairwise comparisons with other algorithms, COLA consistently achieves significantly higher ED than other methods. Additionally, we also observe that as the number of objectives increases, such as in HalfCheetah-5d and Hopper-5d, COLA exhibits more substantial performance improvements compared to other general policy methods. This indicates that COLA exhibits better adaptability to tasks with more objectives. Besides, we present the training process of COLA in comparison to other general policy methods in Figure 3. Notably, COLA demonstrates significant superiority over other algorithms in both HV and UT metrics with an equivalent number of samples, showcasing its higher sample efficiency. Furthermore, we compare the Pareto fronts constructed by different algorithms, which offers a more intuitive representation of the quality of the discovered Pareto fronts. As depicted in Figure 4, COLA constructs a broader and denser Pareto front compared to other algorithms. In the Ant-2d task, COLA's Pareto front is significantly superior to other algorithms. Moreover, in Hopper-3d, COLA almost covers other algorithms. These visualizations consistently highlight the superiority of COLA.

In addition to the main experiments described above, we further evaluate COLA on three additional benchmarks to examine its generality and effectiveness. (1) First, we consider the DeepMind Control Suite (DMC) [34] and extend it to the multi-objective setting, which involves image-based tasks. The objectives of these tasks include energy consumption and the original task goals, such as running speed, walking speed, and jumping height. In this setting, we combined COLA with RAD and compared it against Envelope-RAD with 800,000 environment steps. The results shown in Table 2 indicate that COLA consistently provides significant performance improvements across different tasks. (2) The second benchmark is the continuous-control DeepSea Treasure task proposed in Q-Pensieve [20]. Here, COLA is integrated with SAC and compared against Envelope. As shown in Table 3, COLA achieves better performance on both HV and UT metrics. (3) The third experiment involves two classic tasks from MO-Gym [3], namely Mo-lunar-lander and Mo-mountain-car. As reported in Table 4, we compared against GPI/LS, GPI/PD [35], and MORL/D [36]. The results demonstrate that COLA exhibits a notable performance advantage on Mo-lunar-lander, while on Mo-mountain-car, it shows a slight disadvantage in UT compared to GPI/PD. Overall, COLA demonstrates strong and consistent performance across diverse tasks and algorithmic combinations.

## 4.3 Superiority of Components & Parameter Analysis

In this section, we delve into the impact of COR loss on value function approximation. Our analysis is conducted on MO-Hopper-3d, encompassing three objectives: forward speed, jump height, and energy efficiency. We gather 10,000 states from the interaction between policies and the environment under diverse preferences, calculate the corresponding Monte Carlo ground truth, and determine

Table 4: Experiment on the two tasks from MO-Gym with 10,000 environment steps.

| Task | Metric | GPI/LS | GPI/PD | MORL/D | COLA |
|------|--------|--------|--------|--------|------|
| Mo-lunar-lander | HV ($\times 10^8$) | 4.37 | 4.66 | 3.84 | **6.42** |
| | UT | 13.52 | 15.12 | 3.71 | **30.18** |
| Mo-mountain-car | HV ($\times 10^3$) | 1.79 | **3.46** | 1.10 | 3.31 |
| | UT | -48.57 | **-42.47** | -50.23 | -50.42 |

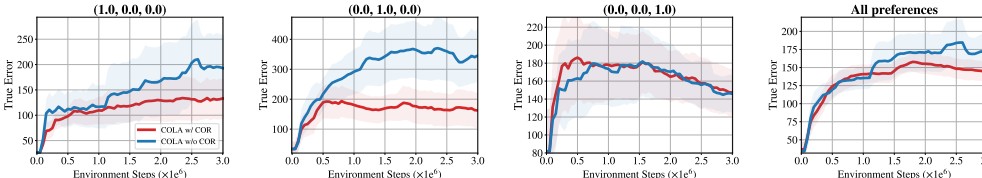

Figure 5: The comparison of ground truth error. The first three plots illustrate the error in three extreme preference settings, while the last plot showcases the average true error across multiple preferences sampled uniformly from the entire preference space.

the errors between the estimated values based on these states and the ground truth. The above follows the evaluation method for value approximation accuracy in TD3 [37]. The experimental results illustrated in Figure 5 demonstrate that using COR results in significantly smaller errors compared to not using it for extreme preferences $(1.0, 0.0, 0.0)$ and $(0.0, 1.0, 0.0)$, while the errors are comparable for $(0.0, 0.0, 1.0)$. Furthermore, we uniformly sample all possible preferences with a step size of $0.25$ to calculate the mean errors. The results show that using COR can lead to smaller mean errors in the value function, indicating that COR is more conducive to maintaining value functions with multiple preferences.

Next, we conduct ablation experiments on two components of COLA, i.e., OADM and COR. We conduct ablation experiments on the Ant task with three objectives and the Hopper task with five objectives. We gradually remove two components. The results in Figure 6 demonstrate that removing COR significantly decreases both UT and HV. Further removal of OADM on top of that leads to a further decrease in the performance of both metrics. This indicates the crucial role of the two components in COLA.

Subsequently, we conduct the hyperparameters analysis introduced by COLA, i.e., $c$ and $\beta$, where $c$ is used to control the conflict boundary, and $\beta$ is used to control the weight of COR. For hyperparameters $c$, we primarily chose values from $\{0.0, 0.25\}$. We conduct the experiments on the Ant-3d task. The results in Figure 7 show that $c = 0.25$ outperforms $c = 0.0$ in terms of both metrics. In experiments, we set $c$ to 0.25 for all tasks. Regarding $\beta$, we select values from $\{1.0, 0.1, 0.01, 0.001\}$. We observe that $\beta = 1.0$ is more efficient, allowing COR to have a greater impact. Thus we set $\beta$ to 1.0 in most multi-objective tasks. For more details, please refer to Appendix B.2. In addition to the experiments above, we provide further analyses in Appendix D, including studies on the number of sampled preferences and the ablations on $s$ and $z$, and a more detailed analysis of the parameter $c$.

## 5 Related Works

The MORL algorithms can be categorized into three categories. Single-policy methods transform multi-objective problems into single-objective problems by using a scalarization function [6, 8, 9]. These methods typically require predefined preferences and cannot be generalized to other preferences. Multi-policy methods approximate the Pareto front of the optimal solution by constructing a set of policies [10–12, 38–40, 4, 35, 3]. These methods commonly face the challenge of high sample complexity, as they explicitly learn each individual policy. PGMORL [4] employs a model to predict the improvement along with preferences to guide the learning process. General policy methods take preferences as input and aim to cover all policies through a single policy network. CN [13] extends single-objective DQN to multi-objective for approximating Q values under different preferences. Besides, CN introduces Diverse Experience Replay to improve sample efficiency. Envelope Q-Learning [14] introduces envelope Q-functions, enabling efficient knowledge sharing

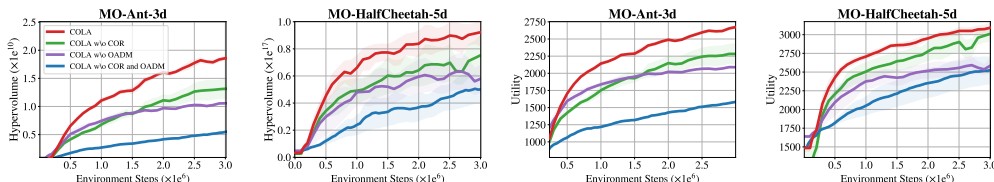

Figure 6: Ablation study on the components of COLA.

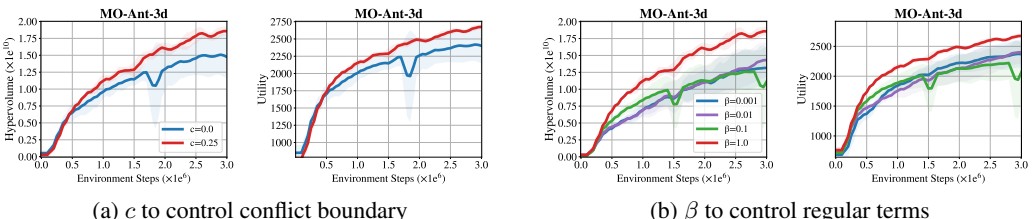

(a) $c$ to control conflict boundary        (b) $\beta$ to control regular terms

Figure 7: Analysis of main hyperparameters.

when learning different preference objectives. PD-MORL [15] guides multi-objective training based on the correlation between preference and Q values. In addition, PD-MORL further enhances performance through parallel sampling and Hindsight Experience Replay [41]. Q-PENSIEVE [20] maintains multiple Q values generated during the learning process to efficiently leverage existing knowledge. CAPQL [33] addresses the failure of linear scalarization methods in multi-objective training by adding a concave term to the immediate reward. Moreover, some works have been devoted to developing benchmark toolkits for MORL [3]. However, existing policy methods face challenges associated with *Inefficiency of Knowledge Sharing* and *Conflict of Optimization Directions*, limiting sample efficiency and ultimate performance. Thus we propose OADM and COR loss to solve the problems.

## 6 Conclusion

In this paper, we propose a novel framework, COLA, to more efficiently cover the entire preference space through a single network. We attribute the primary challenges of current general policy methods to the *Inefficiency of Knowledge Sharing* and *Conflict of Optimization Directions*. To address these challenges, we first propose the Objective-agnostic Latent Dynamics Model. This model explicitly extracts and shares consistent representations in dynamic transitions, establishing a latent space to accelerate multi-objective learning. Additionally, we propose Conflict Objective Regularization to alleviate negative influences among different preferences in the value approximation process, constructing a more accurate value function. In our experiments, we demonstrate the significant superiority of COLA compared with various strong baselines.

## Acknowledgments

This work is supported by the National Key Research and Development Program of China (Grant No. 2024YFE0210900), the National Natural Science Foundation of China (Grant Nos. 624B2101, 62422605, 92370132), and Xiaomi Young Talents Program of Xiaomi Foundation. We would like to thank all the anonymous reviewers for their valuable comments and constructive suggestions, which have greatly improved the quality of this paper.

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

# A Limitations & Future Work

For limitations and future work, firstly our work is empirical proof of the effectiveness of the COLA idea and we provide no theory on optimality, convergence, and complexity. Second, one critical point is the expressivity of the representation method, which is not optimally addressed by this work. Thirdly, our work currently focuses solely on linear utility functions. Therefore, the theoretical foundation, improved representation methods, and the incorporation of nonlinear utility functions are all worthy of further investigation and exploration, which we leave for future work.

In addition, in our previous analysis, we observed a certain degree of policy degradation and forgetting, where behaviors learned in the early stages could not be retained. However, such phenomena have not been observed in population-based multi-objective algorithms. Recently, the integration of evolutionary algorithms (EAs) and RL has been proven effective in single-objective optimization tasks [42–49], but its framework for multi-objective optimization remains underexplored. We believe that exploring this direction, which emphasizes collaborative optimization and the mutual enhancement between RL and EAs [50], can better address the current challenges faced by MORL. We regard this as a key direction for future research in MORL and plan to further explore it based on the COLA framework.

# B Environment Setup

We conduct performance tests in multi-objective MUJOCO tasks. Below are the specific configuration details for each task.

- **MO-HalfCheetah-2d:** Observation and action space dimensionality: $\mathcal{S} \in \mathbb{R}^{17}, \mathcal{A} \in \mathbb{R}^6$, and the environment runs for 500 steps. The first objective is forward speed: $R_1 = v_x + C$. The second objective is control cost: $R_2 = 4 - \sum_i a_i^2 + C$. $C = 1$ is the alive bonus, $v_x$ is the speed in $x$ direction, $a_i$ is the action of each actuator

- **MO-Hopper-2d:** Observation and action space dimensionality: $\mathcal{S} \in \mathbb{R}^{11}, \mathcal{A} \in \mathbb{R}^3$, and the environment runs for 500 steps. The first objective is forward speed: $R_1 = 1.5v_x + C$. The second objective is jumping height: $R_2 = 12 \times (h - h_{\texttt{init}}) + C$. $C = 1 - 0.0002 \times \sum_i a_i^2$ is composed of alive bonus and energy efficiency, $v_x$ is the speed in $x$ direction, $h$ is the current height, $h_{\texttt{init}}$ is the initial height, $a_i$ is the action of each actuator.

- **MO-Walker-2d:** Observation and action space dimensionality: $\mathcal{S} \in \mathbb{R}^{17}, \mathcal{A} \in \mathbb{R}^6$, and the environment runs for 500 steps. The first objective is forward speed: $R_1 = v_x + C$. The second objective is energy efficiency: $R_2 = 4 - \sum_i a_i^2 + C$. $C = 1$ is the alive bonus, $v_x$ is the speed in $x$ direction, $a_i$ is the action of each actuator.

- **MO-Ant-2d:** Observation and action space dimensionality: $\mathcal{S} \in \mathbb{R}^{27}, \mathcal{A} \in \mathbb{R}^8$, and the environment runs for 500 steps. The first objective is x-axis speed: $R_1 = v_x + C$. The second objective is y-axis speed: $R_2 = v_y + C$. $C = 1 - 0.5 \times \sum_i a_i^2$ is composed of alive bonus and energy efficiency, $v_x$ is x-axis speed, $v_y$ is y-axis speed, $a_i$ is the action of each actuator.

- **MO-Hopper-3d:** Observation and action space dimensionality: $\mathcal{S} \in \mathbb{R}^{11}, \mathcal{A} \in \mathbb{R}^3$, and the environment runs for 500 steps. The first objective is forward speed: $R_1 = 1.5v_x + C$. The second objective is jumping height: $R_2 = 12 \times (h - h_{\texttt{init}}) + C$. The third objective is energy efficiency: $R_3 = 4 - \sum_i a_i^2 + C$. $C = 1$ is the alive bonus, $v_x$ is the speed in $x$ direction, $h$ is the current height, $h_{\texttt{init}}$ is the initial height, $a_i$ is the action of each actuator.

- **MO-Ant-3d:** Observation and action space dimensionality: $\mathcal{S} \in \mathbb{R}^{27}, \mathcal{A} \in \mathbb{R}^8$, and the environment runs for 500 steps. The first objective is x-axis speed: $R_1 = v_x$. The second objective is y-axis speed: $R_2 = v_y$. The third objective is energy efficiency: $R_3 = 4 - 0.5 \times \sum_i a_i^2$. $C = 1 - 0.5 \times \sum_i a_i^2$ is composed of alive bonus and energy efficiency, $v_x$ is x-axis speed, $v_y$ is y-axis speed, $a_i$ is the action of each actuator.

- **MO-Ant-4d:** Observation and action space dimensionality: $\mathcal{S} \in \mathbb{R}^{27}, \mathcal{A} \in \mathbb{R}^8$, and the environment runs for 500 steps. Each objective is energy efficiency of one leg and x-axis speed: $R_{\{1,2,3,4\}} = 4.0 - 2 \times (a_{i,0}^2 + a_{i,1}^2) + v_x$. $a_{i,0}$ and $a_{i,1}$ are the action of each actuator on the ith leg. The objective of this task is to enable the agent to maximize velocity

while simultaneously controlling the energy cost of different legs, aiming to achieve precise control of each leg.

- **MO-HalfCheetah-5d:** Observation and action space dimensionality: $\mathcal{S} \in \mathbb{R}^{17}, \mathcal{A} \in \mathbb{R}^6$, and the environment runs for 500 steps. The first objective is forward speed: $R_1 = 0.5 \times v_x + C$. The second objective is jumping height: $R_2 = 20 \times (h - h_{\texttt{init}}) + C$. The third objective is the control cost of the front leg: $R_3 = 4 - 1.5 \times (a_{f,0}^2 + a_{f,1}^2) + C$. The fourth objective is the control cost of the torso: $R_4 = 4 - 1.5 \times (a_{t,0}^2 + a_{t,1}^2) + C$. The fifth objective is the control cost of the hind leg: $R_5 = 4 - 1.5 \times (a_{h,0}^2 + a_{h,1}^2) + C$. $C = 1$ is the alive bonus, $v_x$ is the speed in $x$ direction, $a_{f,0}, a_{f,1}, a_{t,0}, a_{t,1}, a_{h,0}$ and $a_{h,1}$ are the actions of different actuators.

- **MO-Hopper-5d:** Observation and action space dimensionality: $\mathcal{S} \in \mathbb{R}^{11}, \mathcal{A} \in \mathbb{R}^3$, and the environment runs for 500 steps. The first objective is forward speed: $R_1 = 1.5 \times v_x + C$. The second objective is jumping height: $R_2 = 12 \times (h - h_{\texttt{init}}) + C$. The third objective is the energy efficiency of the first actuator: $R_3 = 4 - 3 \times a_0^2 + C$. The fourth objective is the energy efficiency of the second actuator: $R_4 = 4 - 3 \times a_1^2 + C$. The fifth objective is the energy efficiency of the third actuator: $R_5 = 4 - 3 \times a_2^2 + C$. $C = 1$ is the alive bonus, $v_x$ is the speed in $x$ direction, $h$ is the current height, $h_{\texttt{init}}$ is the initial height, $a_i$ is the action of each actuator.

- **MO-Ant-5d:** Observation and action space dimensionality: $\mathcal{S} \in \mathbb{R}^{27}, \mathcal{A} \in \mathbb{R}^8$, and the environment runs for 500 steps. The first objective is x-axis speed: $R_1 = v_x$. The second to the fifth objectives are the energy efficiency of different legs: $R_{\{2,3,4,5\}} = 4.0 - 2 \times (a_{i,0}^2 + a_{i,1}^2) + v_x$. $a_{i,0}$ and $a_{i,1}$ are the actions of each actuator on the ith leg. This task is similar to MO-Ant-4d, but the key difference lies in the desire to achieve precise energy consumption control for different legs under varying speeds.

## B.1 Experiment Details

During training, we predefined a preference set to record HV and UT every 50000 environment steps. These preferences are sampled at regular intervals in the preference space, covering all possible preferences. For two-objective tasks, the preference interval is set to 0.005. For three-objective tasks, the preference interval is set to 0.05. For four-objective tasks, the preference interval is set to 0.1. For five-objective tasks, the preference interval is set to 0.2. Then we use these preferences as conditions for the general policy methods, collecting return vectors for each preference. Based on these vectors, we calculate HV and UT. For algorithms like PGMORL which are multi-policy methods, we select the best policy for each preference to calculate HV and UT. To visualize the Pareto frontier, PGMORL directly plots the discovered set of policies. In contrast, for general policy methods, we set small intervals to explore their potential as much as possible. For two-objective tasks, a preference interval of 0.0001 is used, and for three-objective tasks, a preference interval of 0.002 is employed. All experiments are run on an Intel Xeon E5-2680 v4 @ 2.40GHz machine with an NVIDIA GTX 2080 Ti. Except for the DMC experiments, all other experiments are conducted using only the CPU.

## B.2 Experiments Compute Resources

All experiments are carried out on CPU E5-2680 v4 @ 2.40GHz with 256GB memory

# C Implementation Details

We introduce the implementation details of COLA in this section:

- The baselines we compared include PGMORL[1], Envelope SAC, CAPQL[2], and Q-Pensieve[3]. We reproduce the results based on their official implementations, fine-tuning the hyperparameters involved in each algorithm to achieve the best performance results.

---

[1]https://github.com/mit-gfx/PGMORL
[2]https://github.com/haoyelu/CAPQL
[3]https://github.com/NYCU-RL-Bandits-Lab/Q-Pensieve

Table 5: Details of the hyperparameters of COLA that are varied across tasks.

| Env name | $\beta$ |
|---|---|
| MO-HalfCheetah-2d | 0.01 |
| MO-Hopper-2d | 1.0 |
| MO-Walker-2d | 1.0 |
| MO-Ant-2d | 0.001 |
| MO-Hopper-3d | 1.0 |
| MO-Ant-3d | 1.0 |
| MO-Ant-4d | 1.0 |
| MO-HalfCheetah-5d | 1.0 |
| MO-Hopper-5d | 0.01 |
| MO-Ant-5d | 1.0 |

- We built our code following the official implementation of Q-Pensieve, with the exception that the policy network has layer sizes of $\{128, 128\}$ while keeping the rest of the architecture and hyperparameters consistent.

- For the Objective-agnostic Latent Dynamics Model, a neural network with two layers, each having 256 units, is constructed. It utilizes the ReLU activation function, and the output dimension is set to 50.

- We set $c = 0.25$ for all tasks. Besides, we list the hyperparameters specific to COLA which varied across tasks in Table 5. **if you don't want to tune hyperparameters**, competitive performance can be obtained when keeping $c = 0.25$ and $\beta = 1.0$ in most tasks.

## D   Additional Experiments

Table 6: Comparison between COLA with and without state variable $s$ under different environment steps.

| Env steps | 500k | 1000k | 1500k | 2000k |
|---|---|---|---|---|
| COLA w/ $s$ | **15986288 | 3764** | **16765694 | 3822** | **18089111 | 3996** | **18933218 | 4062** |
| COLA w/o $s$ | 13874916 | 3685 | 16473696 | 3791 | 16564312 | 3870 | 14904042 | 3535 |

**Why do we need to retain $s$ as an input?** This is primarily based on empirical observations. We find that omitting $s$ causes performance fluctuations in some tasks, such as Hopper-2d. The HV and UT results are shown in Table 6: We observe that excluding $s$ leads to performance drops in learning process. Moreover, using $s$ as input is more efficient.

Table 7: Comparison between COLA and COLA with multi-preference ($w$) under 2M environment steps.

| 2M env steps | Walker-2d | Hopper-3d | Ant-2d | HalfCheetah-2d |
|---|---|---|---|---|
| COLA | **4806728 | 2006** | 29541776029 | 3063 | **8722546 | 2696** | 7390226 | 2766 |
| COLA w/ Multi $w$ | 4529242 | 1942 | **29957011493 | 3083** | 7927880 | 2545 | 7221848 | **2798** |

**Is sampling a single preference at each iteration sufficient, or should multiple preferences be sampled instead?** To answer this question, we follow CAPQL and sample a batch of preferences at each step. The results are shown in Table 7. Overall, we find that sampling a single preference yields better performance in most tasks. This aligns with recent findings suggesting that multiple training iterations on the same batch lead to improved outcomes. Fully learning under one preference before switching to another appears more beneficial for training.

**Ablation study on $z$.** We conduct an ablation study using only $c$ while removing $z$. The experimental results are shown in Table 8. We observe that incorporating $z$ leads to better performance compared to the variant without $z$. This improvement is primarily due to the fact that the encoder

Table 8: Comparison between COLA with and without latent variable $z$ under 2M environment steps.

| 2M env steps | Walker-2d | Hopper-3d | Ant-2d | HalfCheetah-2d |
|---|---|---|---|---|
| COLA w/ z | **4806728 \| 2006** | **29541776029 \| 3063** | **8722546 \| 2696** | **7390226 \| 2766** |
| COLA w/o z | 4682616 \| 1992 | 29431977661 \| 3083 | 7974809 \| 2543 | 7036496 \| 2656 |

extracts higher-level latent information from the original state $s$, which helps accelerate the approximation of the critic function and thereby improves learning efficiency.

Table 9: Parameter analysis on the coefficient $c$. The best result for each task and metric is highlighted in **bold**.

| Task | Metric | $c = 0.0$ | $c = 0.25$ | $c = 0.5$ | $c = 1.0$ |
|---|---|---|---|---|---|
| Ant-3d | HV (1e10) | 1.33 | **1.59** | 1.56 | 1.37 |
| | UT | 2285.39 | **2483.19** | 2558 | 2403 |
| HalfCheetah-5d | HV (1e16) | 6.78 | **8.59** | $7.20 \pm 0.81$ | 5.94 |
| | UT | 2962 | **2949** | 2902 | 2939 |

**Analysis of the parameter on $c$.** In all experiments, $c$ is set to 0.25. A smaller value of $c$ indicates that the COR loss has less influence, whereas $c = 1.0$ means that COR is always active, regardless of whether conflicts are present or not. We initially conduct a preliminary comparison between $c = 0.0$ and $c = 0.25$ on a single case and find that $c = 0.25$ yields the best performance. Based on this observation, we adopt $c = 0.25$ consistently across all experiments. We also conduct an ablation study on the effect of $c$, as shown below in Table 9. The results indicate that $c = 0.25$ generally delivers strong performance across different tasks.

