# OpenReview forum: "COLA: Towards Efficient Multi-Objective Reinforcement Learning with Conflict Objective Regularization in Latent Space"
_NeurIPS.cc/2025/Conference — NeurIPS 2025 poster_

### Official Review · Reviewer_4KPM · 2025-06-24

**Clarity:** 3
**Significance:** 3
**Originality:** 3
**Rating:** 4
**Confidence:** 4

**Summary:**

This paper introduces a novel MORL algorithm which provides two big improvements: knowledge sharing and resolution of conflicting gradients. The first improvement is done by sharing knowledge for multiple preferences under a single (conditioned) network and learning a latent model for the dynamics of the environment—this latent code is used as an additional input for both actor and critic. The second improvement is done by adding a loss term (regularization) which prevents the network from forgetting other preferences when updating for a given preference. Some experiments are run on MuJoCo-based environments, tested against some MORL algorithms, and showing the proposed method does well in terms of hypervolume, expected utility, and episodic dominance. There is also an ablation study showing that the two proposed improvements are actually helpful in terms of performance.

**Questions:**

- Why limitting the analysis only to Mujocos? Why not using the other MO-Gymnasium problems?
- Why not comparing the SOTA algorithms: GPI, MORL/D or CMORL?
- How was the hyperparameter search done?
- Did you run experiments where you condition only with `c` (not `z` ) to see if z is adding anything at all? I suspect it should not, since it is essentially an encoded `s`?

**Ethical Concerns:**

["NO or VERY MINOR ethics concerns only"]

**Final Justification:**

My main concern was the lack of experiments and confidence intervals. The authors provided more results and I am now more convinced that the claims are correct.

**Limitations:**

Yes

**Paper Formatting Concerns:**

All good.

**Quality:**

3

**Strengths And Weaknesses:**

## Quality
The paper seems technically sound. I have a few comments that I believe would improve the overall quality of the paper.
1. I like the idea of the gradient resolution. However, I’m missing a bit more intuition on why we need the latent encodings at all, as in the end the raw states are used as input to the models—is the benefit of the latent encoder only coming from `c`, or `z` is playing a role too?
2. This is probably the biggest point: although the paper claims it is comparing against the current SOTA MORL, I believe GPI-LS [2] and MORL/D [3] or even C-MORL [4]—which are the current SOTA—are missing in this.
3. I think the paper could benefit from a bit more experiments: on different environments than Mujoco—MO-Gymnasium provides some. Of course, a few more seeds would help make sure the results are more robust (currently 5) [1].
4. Could the author give insights on how much more compute is needed when adding the improvements? A plot of hypervolume vs. walltime comparing several methods would help understanding how computationally heavy the proposed method is vs. the SOTA.


## Clarity
Overall, the paper is well written and well organized. I just have a few remarks here.

1. For the COR loss (line 184): how is this different from Envelope in the end? It might be nice to have the two losses side by side in the appendix to clearly be able to see where they differ.
2. How did you conduct the hyperparameter search? What was the budget? What was the search space? Optimizing which metric?
3. To what kind of latent size are you compressing the states (z)? And what is the size of c?
4. Fig 2: I think `s_t` is missing as input in the bottom model.
5. Line 125: There is a typo or missing word before “predominantly”.
6. Lines 154-155 add nothing. I’d remove it.
7. Fig 3. The y-axis below should be “expected utility.”
8. Please consider adding the values of the hyperparameters in the appendix for reproducibility.
9. I like that the authors clearly identify the problems they want to tackle: knowledge sharing (note that [3] also emphasizes on this), and gradient conflicts.

## Significance
The proposed improvements seem promising. I would appreciate seeing a comparison vs. GPI and MORL/D, and on different environments than the Mujocos, so we have a better view and understanding of the algorithm’s limits.

## Originality
I like the idea of solving the gradient conflict and think this is opening up a new way of thinking in MORL. There was Envelope doing this before, but it was not identified as a central problem in the community.



## Bibliography
[1] A. Patterson, S. Neumann, M. White, and A. White, “Empirical Design in Reinforcement Learning,” *Journal of Machine Learning Research*, vol. 25, no. 318, pp. 1–63, 2024, [Online]. Available: http://jmlr.org/papers/v25/23-0183.html

[2] L. N. Alegre, A. L. C. Bazzan, D. M. Roijers, and A. Nowé, “Sample-Efficient Multi-Objective Learning via Generalized Policy Improvement Prioritization,” in *Proc. of the 22nd International Conference on Autonomous Agents and Multiagent Systems*, 2023.

[3] F. Felten, E.-G. Talbi, and G. Danoy, “Multi-Objective Reinforcement Learning Based on Decomposition: A Taxonomy and Framework,” *Journal of Artificial Intelligence Research*, vol. 79, pp. 679–723, Feb. 2024, doi: [10.1613/jair.1.15702](https://doi.org/10.1613/jair.1.15702).

[4] Liu, Ruohong and Pan, Yuxin and Xu, Linjie and Song, Lei and You, Pengcheng and Chen, Yize and Bian, Jiang, “Efficient Discovery of Pareto Front for Multi-Objective Reinforcement Learning”, The Thirteenth International Conference on Learning Representations.

---

> ### Author Rebuttal · Authors · 2025-07-31
>
> We appreciate the reviewer's valuable review and constructive comments, and we would like you to know that your questions provide considerably helpful guidance to improve the quality of our paper.
>
> We will try our best to address each of the concerns and questions raised by the reviewer below:
>
> **1. [Re: How did you conduct the hyperparameter search? What was the budget? What was the search space? Optimizing which metric?]**
>
> I primarily focus on two core metrics: HV and UT. The evaluation budget is typically set to 3 million steps. The search space consists of several hyperparameters:
>
>  - c ∈ {0.25, 0.0}
>  - β ∈ {0.001, 0.01, 0.1, 1.0}
>
> Notably, c = 0.25 is kept consistent across all experiments.
>
> **2. [Re: To what kind of latent size are you compressing the states (z)? And what is the size of c?]**
>
> We set latent size to 50 for both z and c.
>
> **3. [Re: Fig 2: I think s_t is missing as input in the bottom model.]**
>
> Thanks to the reviewer for pointing out this oversight. We will address it in the revised version.
>
> **4. [Re: Please consider adding the values of the hyperparameters in the appendix for reproducibility.]**
>
> We provide the detailed hyperparameter settings in the appendix; see Appendix C and Table 2 for more information. **We will also release the code to ensure reproducibility.**
>
> **5. [Re: I would appreciate seeing a comparison vs. GPI and MORL/D, and on different environments than the Mujocos, so we have a better view and understanding of the algorithm’s limits.]**
>
> We thank the reviewer for the valuable suggestion. We conduct evaluations on two continuous action space tasks from MO-Gym. Both sets of experiments are carried out using 100k environment steps.
>
> |  | GPI/LS | GPI/PD | MORL/D | COLA |
> | --- | --- | --- | --- | --- |
> | Mo-lunar-lander HV(10^8) | 4.37 | 4.66 | 3.84 | 6.42 |
> | UT | 13.52 | 15.12 | 3.71 | 30.18 |
> | Mo-mountain-car HV(10^3) | 1.79 | 3.46 | 1.10 | 3.31 |
> | UT  | -48.57 | -42.47 | -50.23 | -50.42 |
>
> These results indicate that COLA exhibits strong or competitive performance on such tasks.
>
> **6. [Re: Did you run experiments where you condition only with c (not z ) to see if z is adding anything at all? I suspect it should not, since it is essentially an encoded s?]**
>
> To address the reviewer's concern, we conduct an ablation study using only c while removing z.
>
> The experimental results are shown in the table below:
>
>
> | 2M env steps | Walker-2d | Hopper-3d | Ant-2d | Half-2d |
> | --- | --- | --- | --- | --- |
> | COLA w z | 4806728 \| 2006 | 29541776029 \| 3063 | 8722546 \| 2696 | 7390226 \| 2766 |
> | COLA w/o z | 4682616 \|1992 | 29431977661 \| 3083 | 7974809 \| 2543 | 7036496 \| 2656 |
>
> We observe that incorporating z leads to better performance compared to the variant without *z*. This improvement is primarily due to the fact that the encoder extracts higher-level latent information from the original state *s*, which helps accelerate the approximation of the critic function and thereby improves learning efficiency.
>
> **7. [Re: For the COR loss (line 184): how is this different from Envelope in the end? It might be nice to have the two losses side by side in the appendix to clearly be able to see where they differ.]**
>
> The COR loss is not necessarily tied to the Envelope framework and can be applied to any MORL algorithm. It can be directly added to the original value function loss as an auxiliary term.
>
> **8. [Re: I think the paper could benefit from a bit more experiments: on different environments than Mujoco.]**
>
> In addition to the experiments mentioned above, we also conduct further evaluations on two other domains:
>
> - **Further experiments on the DeepMind Control Suite, using image-based inputs (800.000 environment steps and 5 seeds)**
>
> The objectives for these tasks include energy consumption and the original task objectives, such as running speed, walking speed, and jumping height. In this case, we combine COLA with RAD and compared it against Envelope-RAD with 800.000 environment steps. The experimental results are shown as follows:
>
> | Tasks | Metric | Envelope | COLA |
> | --- | --- | --- | --- |
> | Cheetah_Run-2d | HV (1e4) | 8.61 ± 0.02 | 12.71 ± 0.02 |
> | Cheetah_Run-2d | UT | 318.8 ± 29.25 | 415.98 ± 30.51 |
> | Hopper_Hop-2d | HV (1e2) | 3.47 ± 1.60 | 31.59 ± 0.14 |
> | Hopper_Hop-2d | UT | 122.06 ± 1.93 | 128.53 ± 0.78 |
> | Walker_walk-2d | HV (1e5) | 5.89 ± 3.98 | 7.73 ± 3.14 |
> | Walker_walk-2d | UT | 238.42 ± 84.43 | 311.69 ± 59.90 |
> - **Experiment on the DeepSea Treasure task from Q-Pensieve with 150.000 environment steps and 5 seeds**
>
>
> | DeepSea Treasure | Envelope | COLA |
> | --- | --- | --- |
> | HV | 799.9 ± 96.995 | 1028.933 ± 3.365 |
> | UT | 6.323 ± 0.6367 | 7.942 ± 0.07538 |
>
> The experiments above demonstrate that COLA can further improve the performance across both HV and UT, whether in image input tasks or other types of tasks.
>
>
> ---
>
> We hope our replies have addressed the concerns the reviewer posed and shown the improved quality of the paper. **We are always willing to answer any of the reviewer's concerns about our work** and we are looking forward to more inspiring discussions.

---

> > ### Comment · Reviewer_4KPM · 2025-08-01
> > **Answer to rebuttal**
> >
> > Thank you for your answers and the additional experimental results. My concerns have been addressed and I will raise my score.
> >
> > > Hyperparameter search
> >
> > I thought the authors had tuned more parameters, e.g., learning rates, NN architecture, etc. This clarifies it.

---

> > > ### Author Response · Authors · 2025-08-01
> > > **We sincerely appreciate the reviewer's recognition and valuable suggestions.**
> > >
> > > We are pleased to have addressed all of the reviewer’s concerns and sincerely appreciate your recognition and support of our work. The constructive suggestions are greatly helpful in improving the quality of our paper. We will incorporate the discussions and additional experiments into the revised version.
> > >
> > > Regarding the hyperparameters, our work is built upon the Q-Pensive codebase, with other hyperparameter settings kept consistent. We commit to releasing the code to ensure reproducibility.
> > >
> > > **If the reviewers have any remaining concerns that have not been addressed, or if there are any other suggestions to further improve the paper, please feel free to inform us at any time.**
> > >
> > > Thank you again for your thoughtful feedback and the valuable discussions.

---

### Official Review · Reviewer_okx7 · 2025-06-29

**Clarity:** 4
**Significance:** 4
**Originality:** 3
**Rating:** 5
**Confidence:** 3

**Summary:**

The paper proposes an empirical method to perform general policy multiobjective reinforcement learning, that is, the authors train a network that is able to output a policy for any arbitrary linear set of weights. The main novelties introduced by this paper focus on the fact that general policy methods are trained by sampling random sets of weights and sequentially training using them, which often results in conficts between the learning objectives. The authors propose to (1) learn the state transition function separately from the reward function, since the transition function is independent from the reward preferences this transfer smoothly across different training iterations; and (2) introduce a regularization term taking into account how much attrition there is with the previous set of weights.

**Questions:**

- What are the assumptions behind the schedule for the set of weights using in the training so that the regularization proposed works as expexted?

- where is PGMORL's results in Figure 3?

- It's not clear how the pareto front results in Figure 4 are built. Are those results for the fully-trained policies queried for different sets of preferences only after the training is done, or are those results for policies during training? same question for the numeric results provided in Table 1.

- For halfcheetah it seems like CAPQL has discovered some policies that way more efficient for speed forward values of around 2000. This could result in significantly superior policies depending on the scenario (if the application mandates power efficiency should be >=2000, for example, CAPQL's policy would be better. This goes against all other resuts presented, I would expect some explanation of in which conditions this behavior could arise, because it could affect where the new proposed method is better or not.

**Ethical Concerns:**

["NO or VERY MINOR ethics concerns only"]

**Final Justification:**

Strengths outweight the weaknesses in special for the work being widely applicable. I recommend the authors add some comments to the manuscript regarding how the choice of preference sampling strategy might affect the performance.

**Quality:**

3

**Strengths And Weaknesses:**

------
Strenghts

- The contributions from the paper are simple to implement and very logical
- There is prospect of relatively high impact in a subarea of RL, and the proposal is widely applicable across pretty much any multiobjective problem.
- Paper is very clear and well-written.
- The proposal is a clear winner in performance in the experimental evaluation performed (which is limited in some ways, see weaknesses section)
-----------
Weaknesses

- For some specific parts of the paper I have the feeling there wasn't much thought devoted to make the approach very general and applicable across other scenarios beyond the evaluated. IN special, I have the feeling the regularization proposed (Eq 7) might not make sense depending on the sequence of set of weights that are presented during training. The regularization only takes the current and the previous set of weights, so if the previous set of weights is very similar to the current one but they are extremely different from the previous 10 sets, there will probably be the "tug of war" effect the authors were trying to avoid and would very likely result in a reduction of performace. There is no comment regarding this in the paper nor an ablation experiment exploring how different regimes for line 4 of the algorithm (when the random set of weights is chosen) would affect performance (Fig 5 shows only uniform sampling and perhaps not even shown in the best way).

- Also related, all the evaluations presented are done in Mujoco tasks, which effectively means there is just a single evaluation domain, it would be good to have some other different domains included to make sure the gain of performance is not only due artifacts of this domain.

---

> ### Author Rebuttal · Authors · 2025-07-31
>
> We appreciate the reviewer's valuable review and constructive comments, and we would like you to know that your questions provide considerably helpful guidance to improve the quality of our paper.
>
> We will try our best to address each of the concerns and questions raised by the reviewer below:
>
> **1. [Re: There is no comment regarding this in the paper nor an ablation experiment exploring how different regimes for line 4 of the algorithm (when the random set of weights is chosen) would affect performance (Fig 5 shows only uniform sampling and perhaps not even shown in the best way). What are the assumptions behind the schedule for the set of weights using in the training so that the regularization proposed works as expected?]**
>
> We thank the reviewer for the helpful suggestion. In our current implementation, we sample a single preference w_i at each step and perform training based on it. This strategy follows the prior work Q-Pensieve, and we did not modify it in order to ensure a fair comparison.
>
> That said, we agree that more efficient preference sampling strategies may exist. To explore this, we conducted additional experiments using an alternative approach: instead of sampling only one preference vector during training, we sample multiple preferences instead of one preference.
>
> | 2M env steps | Walker-2d | Hopper-3d | Ant-2d | Half-2d |
> | --- | --- | --- | --- | --- |
> | COLA | 4806728 \| 2006 | 29541776029 \| 3063 | 8722546 \| 2696 | 7390226 \| 2766 |
> | COLA w/ Multi w | 4529242 \|1942 | 29957011493 \| 3083 | 7927880 \| 2545 | 7221848 \| 2798 |
>
> Overall, we find that sampling a single preference yields better performance in most tasks. This aligns with recent findings suggesting that multiple training iterations on the same batch lead to improved outcomes. Fully learning under one preference before switching to another appears more beneficial for training.
>
> **2.  [Re: Also related, all the evaluations presented are done in Mujoco tasks, which effectively means there is just a single evaluation domain, it would be good to have some other different domains included to make sure the gain of performance is not only due artifacts of this domain.]**
> - **Further experiments on the DeepMind Control Suite, using image-based inputs (800.000 environment steps and 5 seeds)**
>
> The objectives for these tasks include energy consumption and the original task objectives, such as running speed, walking speed, and jumping height. In this case, we combine COLA with RAD and compared it against Envelope-RAD with 800.000 environment steps. The experimental results are shown as follows:
>
> | Tasks | Metric | Envelope | COLA |
> | --- | --- | --- | --- |
> | Cheetah_Run-2d | HV (1e4) | 8.61 ± 0.02 | 12.71 ± 0.02 |
> | Cheetah_Run-2d | UT | 318.8 ± 29.25 | 415.98 ± 30.51 |
> | Hopper_Hop-2d | HV (1e2) | 3.47 ± 1.60 | 31.59 ± 0.14 |
> | Hopper_Hop-2d | UT | 122.06 ± 1.93 | 128.53 ± 0.78 |
> | Walker_walk-2d | HV (1e5) | 5.89 ± 3.98 | 7.73 ± 3.14 |
> | Walker_walk-2d | UT | 238.42 ± 84.43 | 311.69 ± 59.90 |
> - **Experiment on the DeepSea Treasure task from Q-Pensieve with 150.000 environment steps and 5 seeds**
>
> | DeepSea Treasure | Envelope | COLA |
> | --- | --- | --- |
> | HV | 799.9 ± 96.995 | 1028.933 ± 3.365 |
> | UT | 6.323 ± 0.6367 | 7.942 ± 0.07538 |
> - **Experiment on the two tasks from MO-Gym with 10.000 environment steps and 5 seeds**
>
> |  | GPI/LS | GPI/PD | MORL/D | COLA |
> | --- | --- | --- | --- | --- |
> | Mo-lunar-lander HV(10^8) | 4.37 | 4.66 | 3.84 | 6.42 |
> | UT | 13.52 | 15.12 | 3.71 | 30.18 |
> | Mo-mountain-car HV(10^3) | 1.79 | 3.46 | 1.10 | 3.31 |
> | UT | -48.57 | -42.47 | -50.23 | -50.42 |
>
> The experiments above demonstrate that COLA can further improve the performance across both HV and UT, whether in image input tasks or other types of tasks.
>
> **3. [Re: where is PGMORL's results in Figure 3?]**
>
> We sincerely apologize for overlooking this issue. The comparison table is provided below, and we will update the curves in a revised version.
>
> | Ant-3d (env steps) | 1.6 M | 2 M | 2.5 M | 3 M |
> | --- | --- | --- | --- | --- |
> | PGMORL HV (1e9) | 3.10 | 4.63 | 7.22 | 7.80 |
> | UT (1e3) | 1.49 | 1.63 | 1.83 | 1.90 |
> | COLA HV (1e9) | 12.92 | 15.90 | 18.08 | 18.58 |
> | UT (1e3) | 2.39 | 2.48 | 2.61 | 2.68 |
>
> | Walker2d (env steps) | 1 M | 2 M | 2.5 M | 3 M |
> | --- | --- | --- | --- | --- |
> | PGMORL HV (1e6) | 3.41 | 4.31 | 4.45 | 4.50 |
> | UT (1e3) | 1.73 | 1.89 | 1.92 | 1.94 |
> | COLA HV (1e6) | 4.53 | 4.81 | 4.83 | 4.87 |
> | UT (1e3) | 1.96 | 2.01 | 2.02 | 2.02 |
>
> | Half-5d (env steps) | 1 M | 2 M | 2.5 M | 3 M |
> | --- | --- | --- | --- | --- |
> | PGMORL HV (1e16) | 4.31 | 6.54 | 7.08 | 7.52 |
> | UT (1e3) | 2.36 | 2.59 | 2.63 | 2.64 |
> | COLA HV (1e16) | 7.33 | 8.59 | 8.95 | 9.15 |
> | UT (1e3) | 2.73 | 2.95 | 3.07 | 3.11 |
>
> | Hopper-5d (env steps) | 1.6 M | 2 M | 2.5 M | 3 M |
> | --- | --- | --- | --- | --- |
> | PGMORL HV (1e16) | 8.95 | 8.99 | 9.60 | 9.80 |
> | UT (1e3) | 2.69 | 2.70 | 2.71 | 2.71 |
> | COLA HV (1e16) | 9.65 | 9.93 | 10.02 | 10.41 |
> | UT (1e3) | 2.69 | 2.69 | 2.70 | 2.71 |
> **4. [Re: It's not clear how the pareto front results in Figure 4 are built. Are those results for the fully-trained policies queried for different sets of preferences only after the training is done, or are those results for policies during training? same question for the numeric results provided in Table 1.]**
>
> For all evaluation results, including Pareto front evaluations, we assess the performance **after training is complete**, using the trained policy. The full details of the evaluation procedure are provided in **Appendix B.1**.
>
> **5. [Re: For halfcheetah it seems like CAPQL has discovered some policies that way more efficient for speed forward values of around 2000. This could result in significantly superior policies depending on the scenario (if the application mandates power efficiency should be >=2000, for example, CAPQL's policy would be better. This goes against all other results presented, I would expect some explanation of in which conditions this behavior could arise, because it could affect where the new proposed method is better or not. ]**
>
> In our experiments, we observed that in the HalfCheetah task, COLA initially achieves better performance in speed. However, COLA must maintain a significantly larger Pareto front compared to other methods. During training, this requirement can lead to a phenomenon where performance under certain preferences improves as a result of maximizing cumulative returns, while performance under other preferences degrades over time. We believe this may be attributed to a form of catastrophic forgetting, preventing the algorithm from consistently maintaining the best-found performance across all objectives. This issue may require additional constraint mechanisms to mitigate.
>
> Therefore, if the goal is to satisfy a specific requirement—e.g., power efficiency ≥ 2000—this constraint should be explicitly incorporated into the optimization objective, such as by modifying the reward function accordingly. Doing so typically leads to better and more targeted results.
>
>
> ---
>
> We hope our replies have addressed the concerns the reviewer posed and shown the improved quality of the paper. **We are always willing to answer any of the reviewer's concerns about our work** and we are looking forward to more inspiring discussions.

---

> > ### Comment · Reviewer_okx7 · 2025-08-04
> >
> > Thank you for the authors to address the questions and provide a few additional experiments. I will update my score accordintly.

---

> ### Author Response · Authors · 2025-08-05
> **We sincerely appreciate the reviewer's recognition and valuable suggestions!**
>
> We are very pleased to have addressed all of the reviewers’ concerns and are deeply grateful for your positive evaluation of our work and continued support.
>
> Your thoughtful feedback has been instrumental in improving our work, and we will incorporate all relevant details from the discussions into the revised version.
>
> In addition,  we guarantee that the code will be made publicly available at the time of the paper’s release to ensure reproducibility.
>
> Thank you once again for your valuable time and constructive feedback.

---

### Official Review · Reviewer_7qCH · 2025-06-30

**Clarity:** 3
**Significance:** 2
**Originality:** 2
**Rating:** 3
**Confidence:** 4

**Summary:**

This paper proposes a multi-objective reinforcement learning framework called COLA. It introduces a shared base module to model states and state transitions in latent space, facilitating policy learning across different objectives. Additionally, COLA employs a regularization term on the Q-function to manage conflicts between objectives. Experimental results on multiple multi-objective tasks demonstrate COLA's superior performance compared to baseline methods.

**Questions:**

No more questions.

**Ethical Concerns:**

["NO or VERY MINOR ethics concerns only"]

**Final Justification:**

The authors’ rebuttal primarily addressed the concerns from an empirical perspective. However, I still find the explanation insufficient regarding the specific methodological design and the novelty of the work. Therefore, I intend to maintain my current rating. Nevertheless, after reading the reviews from other reviewers, I’m fully comfortable with the final acceptance of this paper.

**Limitations:**

Yes.

**Paper Formatting Concerns:**

No format issues.

**Quality:**

3

**Strengths And Weaknesses:**

Strengths:

1. The paper is clearly written, effectively presenting most methodological details.
2. The methodology is natural and concise, facilitating reproduction and enabling further comparative research.
3. Experimental results on several tasks are promising, clearly showing COLA’s superior performance compared to multiple baselines.

Weaknesses:

1. The rationale for incorporating the latent state-action representation into the inputs of the value function is unclear. From Eq. (3), it appears that the function $D(a, z)$ estimates the expectation of latent representation of the subsequent state resulting from action $a$. However, the benefit and necessity of including this representation in the Q-function inputs remain unclear and intuitively unconvincing for me.
2. The design of the regularization term seems overly simplistic. Regularizing by sampling a single random preference within each iteration may be inefficient, potentially leading to instability or convergence issues.
3. Methodological novelty is somewhat limited. Both latent space modeling of states and state-action transitions, as well as regularization approaches for managing conflicting objectives, are common designs in ML literature.

---

> ### Author Rebuttal · Authors · 2025-07-31
>
> We appreciate the reviewer's valuable review and constructive comments, and we would like you to know that your questions provide considerably helpful guidance to improve the quality of our paper.
>
> We will try our best to address each of the concerns and questions raised by the reviewer below:
>
> **1. [Re: The rationale for incorporating the latent state-action representation into the inputs of the value function is unclear. From Eq. (3), it appears that the function estimates the expectation of latent representation of the subsequent state resulting from action. However, the benefit and necessity of including this representation in the Q-function inputs remain unclear and intuitively unconvincing for me. ]**
>
> Through OADM, the latent state-action representation *c* captures an important piece of information—dynamic information. In locomotion tasks such as those in MuJoCo, it is difficult to accurately estimate the next state based solely on the current state and action. However, this missing information can often be implicitly captured and supplemented by the latent representation *c*.
>
> Based on the state-action representation *c*, the value network effectively gains access to an implicit prediction of the next state. This can be seen as a one-step model rollout, helping the network better understand how a given state-action pair influences long-term returns under the current preference, and thus produce more accurate value estimates.
>
> To address the reviewer's concern, we conduct ablation studies on c across four tasks mentioned above (5 runs, 2M steps): Envelope (UT 1.0, HV 1.0), COLA (UT 1.29, HV 2.14), and COLA w/o c (UT 1.15, HV 1.37). We observe that removing c reduces performance due to absent temporal information, which hinders future state predictions and value approximation.
>
> **2. [Re: Regularizing by sampling a single random preference within each iteration may be inefficient, potentially leading to instability or convergence issues.]**
>
> To address the reviewer’s concern, we follow CAPQL and sample a batch preferences each step. The results are shown below.
>
> | 2M env steps | Walker-2d | Hopper-3d | Ant-2d | Half-2d |
> | --- | --- | --- | --- | --- |
> | COLA | 4806728 \| 2006 | 29541776029 \| 3063 | 8722546 \| 2696 | 7390226 \| 2766 |
> | COLA w/ Multi w | 4529242 \|1942 | 29957011493 \| 3083 | 7927880 \| 2545 | 7221848 \| 2798 |
>
> Overall, we find that sampling a single preference yields better performance in most tasks. This aligns with recent findings suggesting that multiple training iterations on the same batch lead to improved outcomes. Fully learning under one preference before switching to another appears more beneficial for training.
>
> **3. [Re: Methodological novelty is somewhat limited. Both latent space modeling of states and state-action transitions, as well as regularization approaches for managing conflicting objectives, are common designs in ML literature.]**
>
> COLA targets two key challenges in MORL: the difficulty of knowledge sharing and the presence of conflicting objectives. To address these issues, we propose **OADM** and **COR**. OADM is designed to extract temporal dynamic information, while COR serves to constrain conflicting objectives and approximate the value function. **Our goal is to provide a simple yet effective MORL algorithm, supported by extensive experimental evidence**. In addition to the experiments presented in the main text, we further validate the effectiveness of our approach across several additional domains.
> - **experiments on the DeepMind Control Suite, using image-based inputs (800.000 environment steps and 5 seeds)**
>
> | Tasks | Metric | Envelope | COLA |
> | --- | --- | --- | --- |
> | Cheetah_Run-2d | HV (1e4) | 8.61 ± 0.02 | 12.71 ± 0.02 |
> | Cheetah_Run-2d | UT | 318.8 ± 29.25 | 415.98 ± 30.51 |
> | Hopper_Hop-2d | HV (1e2) | 3.47 ± 1.60 | 31.59 ± 0.14 |
> | Hopper_Hop-2d | UT | 122.06 ± 1.93 | 128.53 ± 0.78 |
> | Walker_walk-2d | HV (1e5) | 5.89 ± 3.98 | 7.73 ± 3.14 |
> | Walker_walk-2d | UT | 238.42 ± 84.43 | 311.69 ± 59.90 |
> - **Experiment on the DeepSea Treasure task from Q-Pensieve with 150.000 environment steps and 5 seeds**
>
> | DeepSea Treasure | Envelope | COLA |
> | --- | --- | --- |
> | HV | 799.9 ± 96.995 | 1028.933 ± 3.365 |
> | UT | 6.323 ± 0.6367 | 7.942 ± 0.07538 |
> - **Experiment on the two tasks from MO-Gym with 10.000 environment steps and 5 seeds**
>
>
> |  | GPI/LS | GPI/PD | MORL/D | COLA |
> | --- | --- | --- | --- | --- |
> | Mo-lunar-lander HV(10^8) | 4.37 | 4.66 | 3.84 | 6.42 |
> | UT | 13.52 | 15.12 | 3.71 | 30.18 |
> | Mo-mountain-car HV(10^3) | 1.79 | 3.46 | 1.10 | 3.31 |
> | UT | -48.57 | -42.47 | -50.23 | -50.42 |
>
> ---
>
> We hope our replies have addressed the concerns the reviewer posed and shown the improved quality of the paper. **We are always willing to answer any of the reviewer's concerns about our work** and we are looking forward to more inspiring discussions.

---

> > ### Comment · Reviewer_7qCH · 2025-08-03
> >
> > Thank you for the responses and additional experimental results.
> >
> > The rebuttal mostly addressed my concerns from an empirical standpoint. However, I still feel insufficiency from a methodological perspective, especially regarding the design of sampling only a single random preference and the innovation of methodology.
> >
> > Nevertheless, I'll considering rasining my score in later reviewer-AC discussions.

---

> ### Author Response · Authors · 2025-08-05
> **We sincerely appreciate the reviewer's recognition and valuable suggestions!**
>
> We are pleased to address the reviewers’ concerns through additional experimental investigations.
>
> We would like to offer the following additional clarifications and responses.
>
> 1. **Regarding a single randomly sampled preference during training**,
>     this design choice was guided by empirical observations indicating more efficient optimization. We hypothesize that this benefit arises from allowing the network to more fully exploit each batch with respect to a single preference, thereby enabling better adaptation to its associated distribution. Similar observations have been made in single-objective RL, where reusing batch data has been shown to improve performance and is supported by convergence guarantees [1].
> 2. **Regarding novelty**,
>     - The idea of temporal representation originally emerged in the context of single-agent RL for better sample efficiency. In this work, we adapt and extend it to address challenges unique to MORL. Specifically, we remove the reward prediction component and detach the return gradient to prevent interference of different preferences, and further enhance the representation space by concatenating state and state-action embeddings. To the best of our knowledge, this is the first successful application of temporal representation learning in the MORL setting.
>     - Regarding the regularization term, our approach is most closely related to prior work in multi-task learning that addresses conflicts between different loss functions by directly modifying gradients. In contrast, we focus on value function approximation in MORL and propose a method that mitigates the adverse effects of conflicting optimization objectives. We achieve this through a simple yet effective regularization term. To the best of our knowledge, the use of regularization for resolving conflicts in value function approximation has not been specifically investigated in MORL.
>
>     Thanks to its above components, COLA is designed to be simple yet effective. We have validated its effectiveness across 16 tasks with 3 different domains.
>
>
> We are committed to releasing the code to ensure the reproducibility of our results.
>
> **If there are any remaining questions regarding the experimental design or other aspects of our work, we would be grateful for the opportunity to further clarify. We sincerely appreciate the reviewers’ time and thoughtful feedback.**
>
> [1]: Off-Policy RL Algorithms Can be Sample-Efficient for Continuous Control via Sample Multiple Reuse.

---

### Official Review · Reviewer_G6fw · 2025-07-02

**Clarity:** 4
**Significance:** 3
**Originality:** 3
**Rating:** 5
**Confidence:** 3

**Summary:**

This paper proposes COLA, a novel framework for Multi-Objective Reinforcement Learning (MORL) that combines two key components: an Objective-Agnostic Latent Dynamics Model (OADM) to enable efficient knowledge sharing across objectives, and Conflict Objective Regularization (COR) to mitigate optimization conflicts in the latent space. Experiments on 10 MuJoCo tasks show notable gains over state-of-the-art baselines.

**Questions:**

1. What is the per-iteration training overhead introduced by COR compared to baselines?
2. Is there a theoretical motivation for choosing c = 0.25? Could it be task-specific? Would an adaptive or learned threshold perform better?
3. How does the computational complexity of COR scale with the number of objectives or preference dimensions?
4. The paper attributes instability from discarding raw states s_t to “real-time updates of OADM” (p.4). Could you provide a deeper analysis or diagnostics to clarify this issue?

**Ethical Concerns:**

["NO or VERY MINOR ethics concerns only"]

**Final Justification:**

All my concerns have been satisfactorily addressed.

**Limitations:**

yes

**Quality:**

3

**Strengths And Weaknesses:**

Strengths:
1. The method tackles two core challenges in MORL—inefficient knowledge sharing and conflicting gradients—through a thoughtful combination of latent dynamics modeling (OADM) and regularization (COR). The guiding idea of “seeking commonalities while accommodating differences” is intuitive and well-motivated.
2. The paper is well-written and easy to understand.

Weaknesses:
1. The threshold choice for detecting conflicting preferences is empirically set without theoretical justification. Although the sensitivity analysis (Fig. 7) suggests this value works better than c = 0, its generality across tasks is unclear.
2. The computational cost of COR is not analyzed. Training time and resource usage are not compared to baselines, leaving performance-efficiency trade-offs ambiguous.

---

> ### Author Rebuttal · Authors · 2025-07-31
>
> We appreciate the reviewer's valuable review and constructive comments, and we would like you to know that your questions provide considerably helpful guidance to improve the quality of our paper.
>
> We will try our best to address each of the concerns and questions raised by the reviewer below:
>
> **1. [Re: What is the per-iteration training overhead introduced by COR compared to baselines?]**
>
> We thank the reviewer for the valuable suggestion. To address the concern, we have conducted the runtime experiments. The results show that COR incurs an additional time overhead of approximately 12.9% with one single CPU.
>
> **2. [Re: Is there a theoretical motivation for choosing c = 0.25? Could it be task-specific? Would an adaptive or learned threshold perform better? ]**
>
> In all experiments, *c* is set to 0.25. A smaller value of *c* indicates that the COR loss has less influence, whereas *c* = 1.0 means that COR is always active, regardless of whether conflicts are present or not. We initially conduct a preliminary comparison between *c* = 0.0 and *c* = 0.25 on a single case and find that *c* = 0.25 yields the best performance. Based on this observation, we adopt *c* = 0.25 consistently across all experiments.
>
> We also conducted an ablation study on the effect of *c*, as shown below. The results indicate that *c* = 0.25 generally delivers strong performance across different tasks.
>
> |  | Metric | c = 0.0 | c = 0.25 | c = 0.5 | c = 1.0 |
> | --- | --- | --- | --- | --- | --- |
> | Ant-3d | HV (1e10) | 1.33 | 1.59 | 1.56 | 1.37 |
> |  | UT | 2285.39 | 2483.19 | 2558 | 2403 |
> | HalfCheetah-5d | HV (1e16) | 6.78 | 8.59 | 7.20 ± 0.81 | 5.94 |
> | HalfCheetah-5 | UT | 2962 | 2949 | 2902 | 2939 |
>
> We have not yet identified a robust solution for the automatic adjustment of *c*. One potential approach is to introduce heuristic methods, such as evolutionary algorithms, to explore and iteratively refine multiple *c* values. However, we have observed that *c* = 0.25 consistently serves as an effective choice across a wide range of tasks. For example, the following two domains, COLA also uses *c* = 0.25, further demonstrating the generality of this parameter choice.
>
> - **experiments on the DeepMind Control Suite, using image-based inputs (800.000 environment steps and 5 seeds)**
>
> | Tasks | Metric | Envelope | COLA |
> | --- | --- | --- | --- |
> | Cheetah_Run-2d | HV (1e4) | 8.61 ± 0.02 | 12.71 ± 0.02 |
> | Cheetah_Run-2d | UT | 318.8 ± 29.25 | 415.98 ± 30.51 |
> | Hopper_Hop-2d | HV (1e2) | 3.47 ± 1.60 | 31.59 ± 0.14 |
> | Hopper_Hop-2d | UT | 122.06 ± 1.93 | 128.53 ± 0.78 |
> | Walker_walk-2d | HV (1e5) | 5.89 ± 3.98 | 7.73 ± 3.14 |
> | Walker_walk-2d | UT | 238.42 ± 84.43 | 311.69 ± 59.90 |
>
> - **Experiment on the DeepSea Treasure task from Q-Pensieve with 150.000 environment steps and 5 seeds**
>
> | DeepSea Treasure | Envelope | COLA |
> | --- | --- | --- |
> | HV | 799.9 ± 96.995 | 1028.933 ± 3.365 |
> | UT | 6.323 ± 0.6367 | 7.942 ± 0.07538 |
>
> - **Experiment on the two tasks from MO-Gym with 10.000 environment steps and 5 seeds**
>
> |  | GPI/LS | GPI/PD | MORL/D | COLA |
> | --- | --- | --- | --- | --- |
> | Mo-lunar-lander HV(10^8) | 4.37 | 4.66 | 3.84 | 6.42 |
> | UT | 13.52 | 15.12 | 3.71 | 30.18 |
> | Mo-mountain-car HV(10^3) | 1.79 | 3.46 | 1.10 | 3.31 |
> | UT | -48.57 | -42.47 | -50.23 | -50.42 |
>
> **3. [Re: How does the computational complexity of COR scale with the number of objectives or preference dimensions?]**
> The complexity of COR remains unchanged as the number of optimization objectives increases. This is because, regardless of how many objectives are involved, the optimization procedure still samples a pair of preference vectors (w_i, w_j) from the preference space for each batch of data and computes the COR loss based on Equation (7). Therefore, the computational complexity does not grow with the number of objectives.
>
> **4. [Re: The paper attributes instability from discarding raw states s_t to “real-time updates of OADM” (p.4). Could you provide a deeper analysis or diagnostics to clarify this issue?]**
>
> This is primarily based on empirical observations. We find that omitting s causes performance fluctuations in some tasks, such as Hopper-2d. The HV and UT results are as follows:
>
> | Env steps | 500k | 1000k | 1500k | 2000k |
> | --- | --- | --- | --- | --- |
> | COLA w/ s | 15986288 \| 3764 | 16765694 \| 3822 | 18089111 \| 3996 | 18933218 \| 4062 |
> | COLA w/o s | 13874916 \| 3685 | 16473696 \| 3791 | 16564312 \| 3870 | 14904042 \| 3535 |
>
> We observe that excluding s leads to performance drops in learning process. Moreover, using s as input is more efficient.
>
>
> ---
>
> We hope our replies have addressed the concerns the reviewer posed and shown the improved quality of the paper. **We are always willing to answer any of the reviewer's concerns about our work** and we are looking forward to more inspiring discussions.

---

> > ### Comment · Reviewer_G6fw · 2025-08-04
> >
> > I appreciate the detailed response from the authors. My concerns are all addressed. I will update the score accordingly.

---

> > > ### Author Response · Authors · 2025-08-05
> > > **Sincerely appreciate your recognition and support!**
> > >
> > > We are pleased to have addressed the reviewer’s concerns and sincerely appreciate your recognition and support of our work. The constructive suggestions are greatly helpful in improving the quality of our paper.
> > >
> > > We will carefully incorporate all manuscript refinements and experiments discussed into the revised version.
> > >
> > > In addition,  we guarantee that the code will be made publicly available at the time of the paper’s release to ensure full reproducibility.
> > >
> > > Thank you again for your thoughtful feedback and the valuable discussions.

---

### Decision · Program_Chairs · 2025-09-17

**Decision:**

Accept (poster)

**Comment:**

This work tackles two core challenges in MORL, inefficient knowledge sharing and conflicting gradients. The approach is through a combination of latent dynamics modeling and regularization.

This is the third time I have reviewed this manuscript. I am happy to see the improvement over the submissions to this extent that all reviewers are positive on the submission. I recommend acceptance.